# Adipose Tissue-Derived Stromal Cells Alter the Mechanical Stability and Viscoelastic Properties of Gelatine Methacryloyl Hydrogels

**DOI:** 10.3390/ijms221810153

**Published:** 2021-09-21

**Authors:** Francisco Drusso Martinez-Garcia, Martine Margaretha Valk, Prashant Kumar Sharma, Janette Kay Burgess, Martin Conrad Harmsen

**Affiliations:** 1Department of Pathology and Medical Biology, University Medical Center Groningen, University of Groningen, 9713 GZ Groningen, The Netherlands; m.m.valk.1@student.rug.nl (M.M.V.); j.k.burgess@umcg.nl (J.K.B.); m.c.harmsen@umcg.nl (M.C.H.); 2W.J. Kolff Research Institute, University Medical Center Groningen, University of Groningen, 9713 AV Groningen, The Netherlands; 3Department of Biomedical Engineering, University Medical Center Groningen, University of Groningen, 9713 AV Groningen, The Netherlands; 4Groningen Research Institute for Asthma and COPD (GRIAC), University Medical Center Groningen, University of Groningen, 9713 GZ Groningen, The Netherlands

**Keywords:** viscoelasticity, adipose tissue-derived stromal cells, GelMA, Maxwell elements, cell-matrix interactions, extracellular matrix

## Abstract

The extracellular matrix provides mechanical cues to cells within it, not just in terms of stiffness (elasticity) but also time-dependent responses to deformation (viscoelasticity). In this work, we determined the viscoelastic transformation of gelatine methacryloyl (GelMA) hydrogels caused by adipose tissue-derived stromal cells (ASCs) through mathematical modelling. GelMA-ASCs combination is of interest to model stem cell-driven repair and to understand cell-biomaterial interactions in 3D environments. Immortalised human ASCs were embedded in 5%, 10%, and 15% (*w*/*v*) GelMA hydrogels and evaluated for 14 d. GelMA had a concentration-dependent increase in stiffness, but cells decreased this stiffness over time, across concentrations. Viscoelastic changes in terms of stress relaxation increased progressively in 5% GelMA, while mathematical Maxwell analysis showed that the relative importance (*R_i_*) of the fastest Maxwell elements increased proportionally. The 10% GelMA only showed differences at 7 d. In contrast, ASCs in 15% GelMA caused slower stress relaxation, increasing the *R_i_* of the slowest Maxwell element. We conclude that GelMA concentration influenced the stiffness and number of Maxwell elements. ASCs changed the percentage stress relaxation and *R_i_* of Maxwell elements transforming hydrogel viscoelasticity into a more fluid environment over time. Overall, 5% GelMA induced the most favourable ASC response.

## 1. Introduction

In tissues, cells reside within the extracellular matrix (ECM). The ECM consists of fibrous proteins such as collagens, that provide mechanical strength and architecture; enmeshed in a water-retaining gel of negatively charged polysaccharides, such as glycosaminoglycans (GAGs) [1,2]. The ECM is commonly modelled in vitro using hydrogels—water-swollen polymeric networks formed by physical interactions, covalent crosslinks or both [3]. Hydrogels can be combined with mesenchymal stem cells (MSCs) for tissue engineering and regenerative medicine (TERM) purposes, including 3D bioprinting and to model cell-matrix interactions in 2D and 3D environments [4,5,6].

Within a 3D environment, MSCs will experience biochemical and biomechanical stimuli. Overall, the ECM regulates cellular behaviour via activation of intracellular signalling pathways [7,8,9]. In response, cells may degrade, deposit, and rearrange matrix components [10]. While hydrogel stiffness and its influence on (stem-)cell behaviour have been extensively investigated, this only considers the elastic response within hydrogel mechanics [7,11,12,13,14,15,16,17]. Recently, it has been recognised that hydrogels, just like organs and tissues, are viscoelastic in nature and exhibit creep and stress relaxation properties [12,18]. A viscoelastic material exhibits instantaneous (elastic) and time-dependent (viscous) strain in response to an applied constant stress, or stress relaxation as a response to applied constant strain [19]. Unlike elastic materials that store energy, a viscoelastic material will dissipate energy as a function of time due to viscous flow. Strain energy dissipation can be quantified in terms of stress relaxation. Thus, viscoelasticity is an inherent property of hydrogels that influences cellular responses [18,19]. 

In 3D culture, cells will initially attach to their surrounding ECM and, to put it simply, will “push and pull” on their adjacent environment. This cell-induced strain will be initially resisted by the matrix (elastic response), but this resistance will decay over time (viscous response) [20]. For example, spreading, proliferation and differentiation of mesenchymal cells such as fibroblasts and MSCs, are enhanced in hydrogels with faster stress relaxation (e.g., alginate) independent of the stiffness profile. This denotes that cells can sense the decrease in opposing forces [20,21]. Furthermore, matrix viscoelastic properties can be influenced by cell-derived ECM, as illustrated by chondrocytes cultured within an alginate hydrogel which increased the viscous dissipation, (i.e., facilitated faster stress relaxation) due to GAG deposition [22]. 

While the examples amassed to date offer invaluable knowledge about the influence of viscoelasticity on cellular processes, the current literature derives mostly from non-(bio)degradable hydrogels and at short time scales [20,22,23]. These materials offer a unidirectional view on how viscoelastic microenvironments influence cells, but not vice-versa. Natural ECMs are degradable, with constant renewal by the cells populating them. Hence, initial viscoelastic properties can change over time mediated by cell-induced ECM modifications [24,25].

In this study, we employed adipose tissue derived stromal cells (ASCs) loaded into gelatine methacryloyl (GelMA) hydrogels. ASCs are multipotent and can differentiate into adipogenic, chondrogenic, and osteogenic phenotypes, among others [26]. In comparison to MSCs from bone marrow, higher numbers of ASCs can be easily obtained via minimally invasive liposuction, making their collection more convenient [27]. GelMA is a semi-synthetic hydrogel that stands out due to its short manufacture time, low cost, and tuneability [28,29,30]. GelMA is biocompatible and biodegradable, as it retains in its backbone collagen cell-integrin binding sequences which allows integrin-mediated cell adhesion [31,32,33]. GelMA synthesis is achieved by modifying gelatine’s NH_2_ side chains with methacrylic anhydride (MA) [34]. This process generates methacryloyl groups that form covalent crosslinks in the presence of a photo-initiator upon light exposure (UV/VIS) [35]. These covalent bonds prevent temperature-driven gel-sol transition seen in unmodified gelatine and renders GelMA suitable for use in cell culture at 37 °C [36]. Combination of GelMA-ASCs is of interest in TERM, where GelMA has usually been chemically tweaked or mixed with hyaluronic acid to drive osteogenic and chondrogenic differentiation of MSCs in a concentration-dependent fashion [37,38,39].

In the literature, data that describe the impact of stromal cells over matrix time-dependent mechanics is scarce. Therefore, in our study, we evaluated the changes in GelMA stiffness (elastic modulus) and viscoelasticity (stress relaxation) caused by embedded ASCs cultured over a period of 14 d. The stress relaxation data were analysed using a generalized Maxwell model. ASC morphology and viability was also assessed. Subsequently, we mathematically modelled the viscoelastic changes caused by ASCs-matrix interaction in GelMA hydrogels in vitro.

## 2. Results

### 2.1. GelMA Functionalisation

Examples of the ^1^H-NMR spectra derived from gelatine and GeIMA batches used in these experiments are shown in Figure 1. Two areas from the spectra (Figure 1a,c) were used to calculate the degree of functionalisation (DoF). Following Equation (1), the DoF of the GelMA batch used in these experiments was 56 ± 0.90%.

### 2.2. Swelling 

The mass swelling ratio of GelMA hydrogels based on Equation (2) is shown in Figure 2. Theoretical swelling ratio at 0 d of 5%, 10%, and 15% *w*/*v* GelMA were 20.00, 10.00, and 6.70, respectively. After reaching equilibrium swelling at 1 d, the calculated swelling ratio were 14.19 ± 1.28, 6.41 ± 0.43, and 4.54 ± 0.46 for 5%, 10%, and 15% GelMA, respectively. Analyses showed that cell-free 5%, 10%, and 15% GelMA hydrogels differed after 1 d with 5% GelMA having the highest water content compared to 10% GelMA and 15% GelMA. Compared to 15% GelMA, 10% GelMA hydrogels also contained more water (Figure 2). 

### 2.3. Elastic Modulus 

The elastic modulus (stiffness) at 0 d of cell-free (control) 5%, 10%, and 15% GelMA hydrogels at a strain rate of 0.2 s^−1^ was 6.94 ± 1.89 kPa; 72.08 ± 9.50 kPa; and 214.50 ± 24.53 kPa, respectively. Stiffness of cell-free GelMA did not change during the 14 d of study. The elastic modulus at 0 d of cell-loaded GelMA hydrogels did not differ from cell-free hydrogels at the same timepoint (Figure 3a). 

In cell-loaded 5% GelMA, the elastic modulus at 0 d (6.39 ± 0.59 kPa), steadily decreased at 7 d (4.96 ± 0.75 kPa) and 14 d (4.86 ± 0.99 kPa). The decrease in stiffness was also detected between cell-loaded and cell-free hydrogels at 1 d, 7 d, and 14 d (Figure 3a). 

The elastic modulus of cell-loaded 10% GelMA at 1 d (70.83 ± 15.32 kPa) decreased after 7 d (64.41 ± 8.66 kPa) and 14 d (68.81 ± 13.43 kPa) (Figure 3a). Differences between cell-loaded and cell-free hydrogels were found at 7 d.

In cell-loaded 15% GelMA hydrogels, their elastic modulus decreased at 14 d (189.90 ± 13.67 kPa) compared to 0 d (217.20 ± 20.74 kPa), 1 d (216.20 ± 21.01 kPa), and 7 d (216.40 ± 22.19 kPa) (Figure 3a). Differences at 14 d between cell-loaded and cell-free hydrogels were also found.

### 2.4. Stress Relaxation 

The percentage of stress relaxation of cell-free 5%, 10%, and 15% GelMA hydrogels in 100 s at 0 d was 7.2 ± 1.48%, 7.34 ± 1.12%, and 6.95 ± 0.64%, respectively. This initial percentage of stress relaxation did not change significantly after 14 d in culture conditions in cell-free hydrogels (Figure 3b). Furthermore, comparisons between GelMA concentrations and time points showed that virtually all cell-free hydrogels exhibited stress relaxation percentage lower than 15% in 100 s.

Cell-loaded 5% GelMA hydrogels showed a gradual and significant increase in stress relaxation percentage. Compared to 0 d (8.19 ± 1.12%), stress relaxation percentage increased at 7 d (10.49 ± 2.02%) and 14 d (11.03 ± 2.20%). Differences were also found between cell-loaded hydrogels between 1 d (8.80 ± 1.02%) and 14 d. Higher stress relaxation percentage among cell-free and cell-loaded hydrogels at 7 d and 14 d was also found (Figure 3b). 

Unlike cell-loaded 5% GelMA, cell-loaded 10% GelMA hydrogels showed higher percentage of stress relaxation only at 7 d (7.77 ± 0.84%) compared to 7 d cell-free hydrogels (6.65 ± 0.61%) (Figure 3b). 

In contrast to previous GelMA concentrations, in 15% GelMA cells decreased the percentage of stress relaxation, from 0 d (8.00 ± 1.37%) to 7 d (6.28 ± 1.34) and 0 d to 14 d (6.46 ± 0.87. Decrease in stress relaxation was also found between 1 d (7.80 ± 0.97) and 7 d and 1 d to 14 d (Figure 3b). The average stress relaxation as a function of time during LLCT can be found in Figure 3c.

### 2.5. Maxwell Analysis

Fitting stress relaxation data into a Maxwell model determined the number of Maxwell elements. According to the average tau (*τ*) values (Table 1), the first Maxwell element remained active during *τ*_1_ ≤ 1 s; the second element 1 < *τ*_2_ ≤ 10 s; the third element 10 < *τ*_3_ ≤ 100 s and the fourth element was active *τ*_4_ >100 s. Based on these criteria, 5% GelMA hydrogels required three Maxwell elements, with no first element detected. In both 10% and 15% GelMA hydrogels all four Maxwell elements were required to fit the experimental data (Figure 4). The presence of cells did not influence the number of Maxwell elements at any timepoint. 

The relative importance (*R_i_*) of certain Maxwell elements behaved similarly across GelMA concentrations irrespective of cells. For example, the *R_i_* of the first and second elements in 10% and 15% GelMA remained below 3% (Figure 4b–e). In all GelMA concentrations and conditions, the highest *R_i_* was found in the Maxwell element with the longest relaxation time constant i.e., the fourth element (Figure 4i–k). 

In 5% GelMA, comparison between 0 d and 14 d showed that cells increased the *R_i_* of both the second and third Maxwell elements, while the *R_i_* of the fourth element decreased in the same period of time. Such differences were also found between cell-loaded and cell-free hydrogels at the timepoints before-mentioned (Figure 4a,f,i).

In 10% GelMA, only the fourth element decreased in *R_i_* between 0 d compared to 14 d and between cell-loaded and cell-free hydrogels at 14 d. All the other Maxwell elements remained unchanged. Differences were also found in the fourth element for cell-free hydrogels between 1 d and 7 d (Figure 4j). 

For cell loaded 15% GelMA hydrogels, the *R_i_* of the third element decreased significantly between 0 d and 7 d; 0 d and 14 d; and 1 d to 7 d. Differences between 7 d cell-loaded and cell-free hydrogels were also found. In contrast, the *R_i_* of the fourth Maxwell element increased between 0 d and 7 d; 1 d to 7 d; 1 d to 14 d. Differences between the *R_i_* of cell-loaded and cell free hydrogels at 7 d were also found. Among cell-free hydrogels, the *R_i_* at 0 d was greater than at 1 d (Figure 4h,k). 

### 2.6. Cell Morphology and Viability

All cells exhibited a rounded morphology immediately after seeding in all GelMA concentrations. For cells cultured in 15% GelMA, this initial morphology did not change after 14 d (Figure 5a) and their viability significantly decreased from 90.1% at 0 d, to 37.46% at 7 d and to 24.34% at 14 d (Figure 5b). In contrast, cells in 5% GelMA started spreading as soon as 1 d and were larger and also stretched through the gel. The ASC in 5% GelMA also had filipodia throughout these experiments (Figure 5a). Median cell viability in 5% GelMA, decreased from 98% at 0 d to 84% after 14 d (Figure 5b). In 10% GelMA, cells spread to a lesser degree, but did display an increased in volume (size) compared to cells in 5% GelMA, with this process being more prominent in areas of clustered cells (Figure 5a). In 10% GelMA, median cell viability decreased from 93% at 0 d to 76% at 14 d (Figure 5b). Nevertheless, cell viability decreases detected in 5% and 10% GelMA were not significant. Cells were visible throughout all the conditions examined and occasionally on top of the gels.

## 3. Discussion

In this study, we found that ASCs modify not only the stiffness but the viscoelasticity of GelMA hydrogels. We showed that cells transformed hydrogel viscoelasticity into a more fluid-like environment. We established that cell-induced viscoelastic changes can be accurately quantified using a generalised Maxwell model of viscoelasticity. Through this model we found that GelMA concentration influenced the elastic modulus (stiffness)and number of Maxwell elements, whereas cells influenced percentage stress relaxation and relative importance (*R_i_*) of individual Maxwell elements over time.

GelMA hydrogels showed a concentration-dependent rise in stiffness. The elastic modulus of our hydrogels, at a strain rate of 0.2 s^−1^, differed from other GelMA hydrogels with a similar polymer concentration and degree of functionalisation (DoF 56%) [30,32,40]. Such variations in stiffness can be due to the wide variety of photopolymerisation conditions. Higher light intensity augments the stiffness of GelMA hydrogels with equal DoF, polymer and photoinitiator concentrations [30]. Since the viscoelastic nature of hydrogels makes their stiffness strain rate-dependent, inconsistencies can be due to the diverse methods used in assessing the elastic properties, as remarked in a recent review [41]. 

Cells did not interfere with GelMA photopolymerisation and crosslink formation, nor was cell viability affected by exposure to UV/VIS-light radiation. Compared to more concentrated hydrogels, 5% GelMA had greater cell viability, marked spreading, and elongated morphology over 14 d. Lower cell viability has also been reported in other hydrogels used for 3D cell culture with increasing polymer concentrations [40]. In our work, embedded cells reduced the stiffness of GelMA hydrogels at varying rates irrespective of polymer concentration. The elastic modulus of 5% GelMA started diminishing within 1 d of cellular activity, while in 10% and 15% GelMA, a decrease in stiffness was detected much later i.e., at 7 d and 14 d, respectively. This corroborates with the reduction of GelMA hydrogel tensile strength by 3T3 murine fibroblasts over a 96 h period [42]. Even so, the elastic modulus found in our work is in the scope of reported physiological stiffness of lung, muscle, and cartilage, corresponding to stiffness of 5%, 10%, and 15% GelMA, respectively [41,43]. The organs and tissues before-mentioned may require tissue-engineered approaches addressing pathological events. Thus, employing GelMA-ASC constructs that resemble the native stiffness of the organ in need of repair may help to improve the design of TERM-aimed constructs.

Viscous dissipation of mechanical stress (stress relaxation) depends on the water content of a hydrogel, i.e., its swelling ratio. Our data showed that the GelMA swelling ratio was inversely proportional to the polymer concentration, which corroborates previous literature [14,40,44,45]. Comparison between the theoretical swelling ratio at 0 d from the calculated at 1 d indicates that GelMA hydrogels lost mass within 24 h. Unlike the elastic modulus, stress relaxation of freshly prepared hydrogels remained unaffected by GelMA concentration and swelling ratio at a value of ~15% in 100 s. In contrast, decellularised ECM (dECM) hydrogels exhibit more than 90% stress relaxation under similar LLCT conditions [43,46,47]. Rheometry data from other authors also demonstrate a predominance of the elastic component of GelMA [48]. Thus, GelMA hydrogels behave as viscoelastic solids whereas dECM hydrogels behave as viscoelastic fluids. The viscoelastic nature of hydrogels depends on the type of bonds and crosslinks that bind the polymer network together. Overall, hydrogels formed by covalent and ionic crosslinks tend to behave as viscoelastic solids, and have lower stress relaxation compared to ECM-derived hydrogels, predominantly bound through physical interactions. [18,25,49,50]. Nevertheless, our data shows that despite the covalent nature of GelMA bonds, their viscoelastic behaviour can change due to cell-matrix interactions.

Maxwell analysis of the viscoelastic properties of GelMA, indicated that the number of Maxwell elements increased alongside stiffness, an observation also made in other hydrogel systems [43,46]. In our study, ASCs cultured in 5% GelMA increased the percentage of stress relaxation, transforming their pericellular elastic environment into a more viscoelastic fluid-like environment. Human bone marrow-derived MSCs caused a similar effect on PEG hydrogels [24]. Matrix viscoelasticity requires a molecular rearrangement to dissipate stress, which in our study was reflected by the changes in the individual Maxwell elements. Viscoelastic modelling of 5% GelMA data showed that the second and third Maxwell elements, i.e., the fastest and intermediate elements, increased their *R_i_* at the cost of a decrease in the fourth element, i.e., longest element. For 10% GelMA, only the fourth element changed after 14 d, but no statistical changes were reflected in other elements, likely due to small changes in their *R_i_*. In contrast to previous concentrations, in 15% GelMA, ASCs slowed down the stress relaxation. Maxwell analysis indicated that the *R_i_* of the fourth element increased, with changes reflected on the third element. These findings suggest that matrix mechanics can lead to diverse viscoelastic outcomes in ASC loaded hydrogels. Cell-induced alterations in the architecture and composition of the ECM may explain the contrasting results in viscoelasticity among GelMA concentrations. For example, deposition of hygroscopic ECM molecules such as GAGs has been associated with increased viscous dissipation [22]. Additionally, hydrogels are porous matrices exhibiting poroelastic relaxation—solvent migration away from stress [51]. Tightly crosslinked hydrogels with reduced spatial architectures can limit such migration, and while cells can modify GelMA porosity [42], the overall impact of such alterations on time-dependent mechanics is elusive. So far, most mathematical models are unable to reconcile poroelasticity from viscoelasticity, as both co-exist in time [52,53]. Furthermore, hydrogels are also viscoplastic, with cells capable of permanently deforming the polymer network, and subsequently changing the mechanical properties [25,54,55]. All of these parameters could play a role in the viscoelastic remodelling of GelMA by ASC but decoupling them from each other is beyond the scope of our current study. Other limitations should be mentioned. In 3D environments, cells are located at different microscopy focal planes which limit tracking changes in morphology or migration patterns from particular hydrogel regions over time. This limitation is pronounced in areas of fluorescently-labelled clustered cells, as their distribution increases the fluorescent signal detected, causing areas to look overexposed during imaging and prevent a proper visualising of the cellular behaviour. LLCT performed at room temperature might underrepresent hydrogel viscoelasticity experienced at a physiological temperature. LLCT measured global changes, but it did not detect microscale changes caused by the cells to their pericellular region. Distinct cell-loading densities and cell viability might also affect hydrogel viscoelasticity, but it is unknown as to what degree it could alter hydrogel time-dependent mechanics. 

## 4. Materials and Methods

### 4.1. GelMA Synthesis

Medical grade gelatine type A, 99.8 kDa MW, 262 g Bloom (MedellaPro^®^ > 600MW, Leverkusen, Germany) was dissolved in 1x Dulbecco´s phosphate-buffered saline (DPBS; BioWhittaker^®^, Walkersville, MD, USA) and 0.6 mL of MAA (Sigma-Aldrich, Darmstadt, Germany) added per gram of gelatine at 60 °C under gentle stirring. After three hours, the solution was diluted in an equal volume of 1× DPBS and centrifuged at 2000× *g* for 5 min to separate any unreacted MAA. The supernatant was collected, and its pH adjusted to 7.4 with 2 M sodium hydroxide (NaOH). The pre-GelMA stock solution was dialysed for a week in a 14 kDa cut-off cellulose membrane (Sigma-Aldrich) against demi-water at 50 °C with twice daily changes of demi-water. The remaining solution was lyophilised in a Free Zone^®^ 2.5 Plus freeze dryer (Labconco Corporation, Kansas City, MO, USA) at −80 °C for five days. The lyophilized GelMA was stored in the dark at −20 °C until further use.

### 4.2. Determination of GelMA Degree of Functionalization (DoF)

The efficiency of substitution of NH_2_ groups by methacryloyl groups, also known as the degree of functionalization (DoF), was assessed through ^1^H-Nuclear Magnetic Resonance (^1^H-NMR) as described previously [56]. For this, 20 mg of both GelMA and non-modified gelatine were dissolved in 1 mL of D_2_O at 40 °C. The samples were placed inside 5 mm diameter, 100 MHz Wilmad^®^ NMR tubes (Wilmad-LabGlass, Vineland, NJ, USA) and analysed with an Avanced^TM^ Neo 600 MHz spectrometer (Bruker, Rheinstetten, Germany). Each sample underwent 16 cycles with a 5 s delay between cycles. The resulting ^1^H-NMR-spectra were analysed using MestReNova^®^ v14.0 (Mestrelab Research S.L., Santiago de Compostela, Spain). The chemical shift scale was adjusted using the D_2_O peak and normalised using phenylalanine’s aromatic rings. The DoF were calculated by dividing the area of the lysine peak before (*b*) and after (*a*) functionalization as mentioned in Equation (1).
(1)$$DoF=1-\left(\frac{\;a\;}{b}\right)\;\times \;100$$

### 4.3. Hydrogel Preparation

Lyophilised GelMA was dissolved in 1× DPBS at 5%, 10%, and 15% (*w*/*v*) concentrations at 60 °C and mixed with lithium phenyl-2,4,6-trimethylbenzoylphosphinate (LAP; Allevi Inc., Philadelphia, PA, USA), a photoinitiator, at 0.5% (*w*/*v*) concentration. Once dissolved, the GelMA working solution was sterilised with a 0.2 µm polyethersulfone (PES) membrane filter (Corning^®^, Kaiserslautern, Germany) and stored at −20 °C. 

### 4.4. 3D Cell Culture and Photopolymerisation

Immortalised human adipose tissue-derived stromal cells (iADSC13) between passages 19 to 23, negative for *Mycoplasma* spp, were expanded in culture medium composed of high glucose Dulbecco’s modified Eagle medium (DMEM; Lonza, Walkersville, USA) containing 10% Fetal Bovine Serum (FBS; Sigma-Aldrich), 1% Penicillin–Streptomycin (Pen-Strep; Gibco^TM^, Paisley, Scotland) and 2 mM L-glutamine (BioWhittaker^®^, Verviers, Belgium) and cultured at 37 °C, 5% CO_2_. Upon reaching 80–90% confluence in tissue culture plates, cells were detached using 0.5% Trypsin EDTA (Gibco) and centrifuged at 2000× *g* for 5 min. The cell pellet was resuspended in fresh culture medium and the number of live cells calculated using both a Z2^TM^ Coulter Counter^®^ particle count and size analyser (Beckman Coulter^TM^, Brea, CA, USA) and by trypan blue staining. Aliquots containing 7 × 10^5^ cells were centrifuged again, the supernatant removed, and the cell pellet resuspended in 350 µL of either 5%, 10%, or 15% GelMA working solution thawed at 37.5 °C. For hydrogel moulding, polycaprolactone (PCL; Allevi Inc.) containers of 2.4 mm (h) × 14 mm (d) were 3D printed under sterile conditions with a Biobots I bioprinter (Allevi Inc.) at 130 °C, 90 PSI. Hydrogels were cast in the PCL moulds and photopolymerised using the bioprinter’s UV lamp (405 nm) at 7 mW/cm^2^ for 5 min. Cell-free GelMA hydrogels were used as controls. After photopolymerisation, all hydrogels were carefully removed from the moulds and incubated at cell culture conditions (37 °C, 5% CO_2_) until subsequent analysis.

### 4.5. Cell Viability 

Cell viability, spreading, and morphological examination of ASCs were assessed using a Live/Dead fluorescent staining at 0 d, 1 d, 7 d, and 14 d. Cell-loaded hydrogels were washed three times with DPBS and mixed with the Live/Dead working solution composed of 5 µM calcein-AM (Life Technologies^®^, Eugene, OR, USA) and 2 µM propidium iodide (PI; Sigma-Aldrich) dissolved in culture medium in the absence of FBS. Hydrogels were gently shaken at room temperature for 10 min to facilitate diffusion of the working solution and then incubated at 37 °C, 5% CO_2_ for an additional 20 min. The hydrogels were washed again with DPBS and imaged with an EVOS^®^ FL digital inverted microscope (Electron Microscopy Sciences, Hatfield, PA, USA) using the following light cubes: GPF (λ_ex_ 470/22 nm/λ_em_ 525/50 nm), and Texas Red (λ_ex_ 585/29 nm/λ_em_ 628/32 nm) to visualise calcein AM and PI, respectively, at 10x magnification. Micrographs were analysed using using ImageJ Ver 1.52p (https://imagej.nih.gov/ij/; Accessed: 12 February 2021) [57]. All images were transformed into 8-bit, threshold and contrast were corrected to reduce hydrogel autofluorescence, which facilitated particle segmentation and quantification. 

### 4.6. Swelling 

The swelling ratio tells us the fluid to solid ratio in the hydrogel, i.e., weight of DPBS present in the hydrogel divided by the dry weight of GelMA. Acellular GelMA hydrogels were incubated in 1× DPBS for 1 d to reach swelling equilibrium. Afterwards, hydrogels were briefly blotted against paper to remove excess liquid drops and weighed (*Ws*) on a balance (Sartorius Lab Instruments, Göttingen, Germany) before being placed inside a RapidVap^®^ vacuum evaporation system (Labconco Corporation). Hydrogels were dried at 45 °C using negative pressure at 25% vortex speed for 1 d. The remaining weight (*Wd*) was measured, and the swelling ratio calculated according to Equation (2).
(2)$$Swelling\;ratio=\frac{Ws-Wd}{Wd}$$

### 4.7. Characterisation of GelMA Viscoelastic Properties

Changes in the mechanical stability and viscoelasticity of cell-loaded and cell-free (control) GelMA hydrogels were monitored at 0 d, 1 d, 7 d, and 14 d, by performing a stress relaxation test in compression on the Low-Load Compression Tester (LLCT) [58,59,60]. Data acquisition and visualisation in real-time was carried out using LabVIEW 7.1, and subsequently analysed using a data fitting routine written in MatLab 2018 (MathWorks^®^ Inc., Natick, MA, USA). At first the hydrogel was compressed to 80% of its original thickness (i.e., a strain, ε, of 0.2 was induced) at a strain rate ($\dot{\varepsilon }$) of 0.2 s^−1^ in an enclosed environment at ≅ 25 °C. All gels were compressed with a 2.5 mm diameter plunger at three different locations, ensuring at least 2.5 mm distance between both the hydrogel edges and each compression site. During compression, the increase in stress was continuously measured and the slope between stress and strain curve was taken as the elastic modulus. Once the strain reached 0.2, it was maintained at this level for 100 s and the stress continuously monitored. Percentage stress relaxation was calculated by comparing the stress at *t* = 0 s and *t* = 100 s. The relaxing stress as a function of time (*σ*(*t*)) was divided by the constant strain of 0.2 to obtain the value of relaxing modulus *E*(*t*). A generalised Maxwell model shown in Equation (3) was then fitted to the experimental data to get the values of *E_i_* and *τ_i_* for individual Maxwell elements. The *τ_i_* was the relaxation time constant for each individual Maxwell element and was the ratio of *η_i_* (dashpot) and *E_i_* (spring) for that element. The number of Maxwell elements necessary to fit the experimental data was determined by both looking at the fit visually and with the help of the plot which shows a decrease in Chi^2^ value with the addition of every extra Maxwell element. The number of Maxwell elements were chosen for which no further decrease in Chi^2^ was observed.
(3)$$E\left(t\right)\;=E_{1}e^{-t/\tau_{1}}\;+\;E_{2}e^{-t/\tau_{2}}\;+\;E_{3}e^{-t/\tau_{3}}\;+\;\ldots \;E_{n}e^{-t/\tau_{n}}$$

The relative importance (*R_i_*) of each Maxwell element in terms of percentage within the relaxation process was expressed as the proportion of its spring constant, *E_i_*, to the sum of all spring constants (Equation (4)) [58,59,60].
(4)$$R_{i}=\;100\;\frac{E_{i}}{\sum_{i=1}^{n}E_{i}}$$

### 4.8. Statistical Analyses

All statistical analyses were performed using GraphPad Prism v9.1.0 (GraphPad Company, San Diego, CA, USA). All data were searched for outliers using the robust regression and outlier removal (ROUT) test and analysed for normality using Shapiro–Wilk and D’Agostino & Pearson tests [61,62,63]. Based on this, LLCT data with an R^2^ value > 0.950 from the linear regression stress–strain curve data were analysed using two-way ANOVA and Tukey’s post-test. Swelling ratio data were analysed with one-way ANOVA and Tukey’s post-test. Cell viability was analysed using Brown–Forsythe and Welch ANOVA and Dunnett´s T3 post-test. Graphs are presented as mean values with standard deviation (SD) or median values with quartiles. p values below 0.05 were considered statistically significant.

## 5. Conclusions

We conclude that ASCs transformed GelMA viscoelasticity into a more fluid environment and altered hydrogel mechanical stability. Out of the concentrations studied here, 5% GelMA provided the most conducive biomechanical environment for ASC cells to spread and remain viable. Their cellular response in turn, caused the most viscoelastic remodelling towards viscoelastic fluid, observed as changes in elastic modulus, stress relaxation, and *R_i_* of individual Maxwell elements over time. Whether changes in the *R_i_* of each Maxwell element are due to matrix degradation, ECM deposition, or cellular processes that induce permanent deformation of the hydrogel structure, remains an intriguing question.

## Figures and Tables

**Figure 1 ijms-22-10153-f001:**
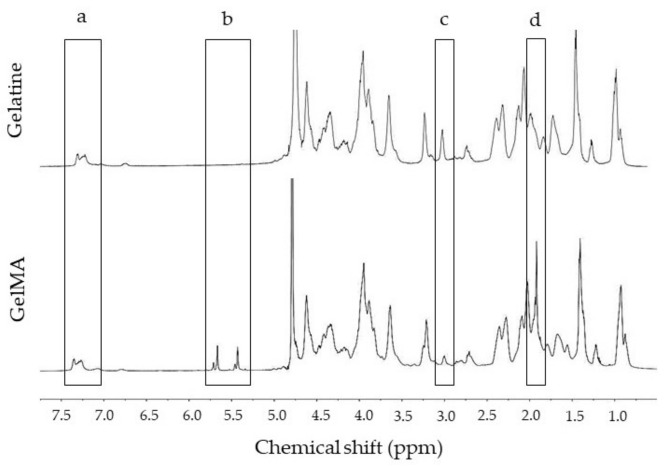
Representative NMR spectra of unmodified gelatine and GelMA. (**a**) Phenylalanine’s aromatic rings were used for normalisation around 7.35 ppm (5 protons). (**b**) Methacryloyl groups in GelMA seen as a double peak between 5.3–6.8 ppm absent in gelatine. (**c**) Lysine and hydroxylysine peaks before and after methacrylation at 2.90 ppm. (**d**) Corresponds to NH_2_ group substitutions by methyl protons of methacryloyl groups absent in unmodified gelatine at 1.8 ppm.

**Figure 2 ijms-22-10153-f002:**
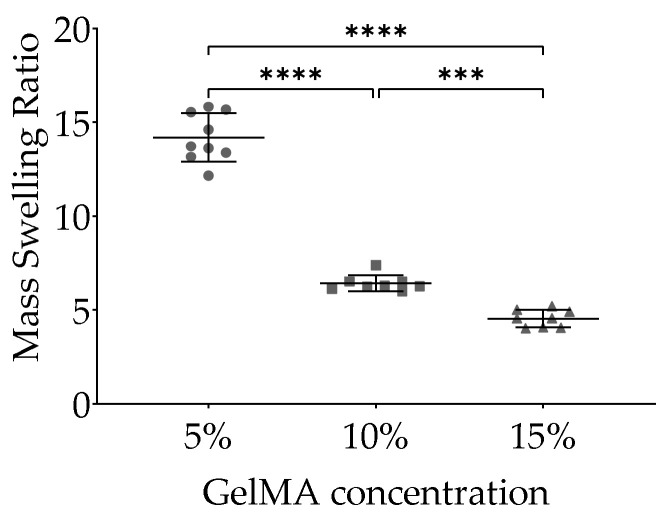
Swelling ratio of GelMA hydrogels. After swelling in DPBS for 1 d and subsequent dehydration the swelling ratio was calculated according to Equation (2). Data derived from three samples per experiment from three independent experiments are shown.. Data are presented as mean with standard deviation. *p* values are indicated *** *p* < 0.001 and **** *p* < 0.0001 according to One-way ANOVA and Tukey.

**Figure 3 ijms-22-10153-f003:**
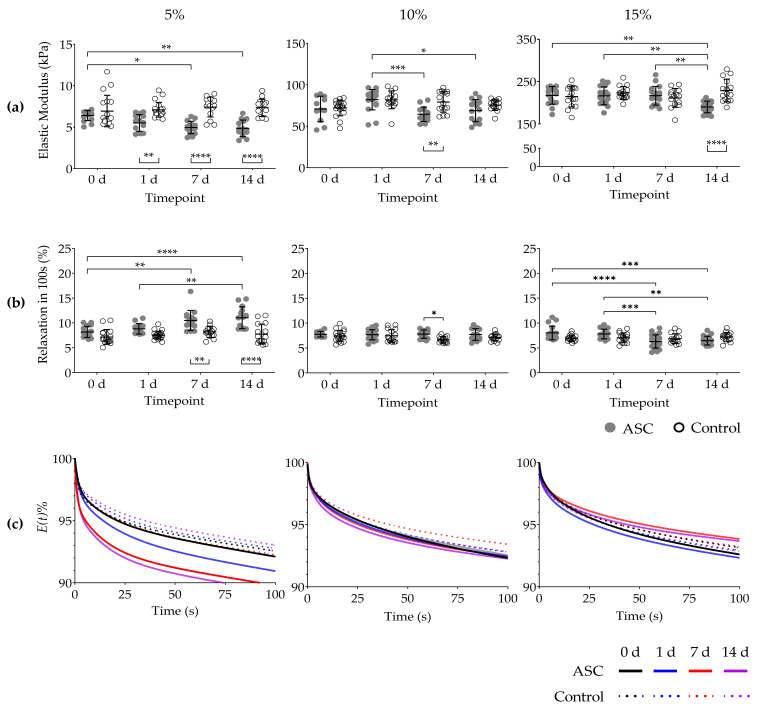
Elastic modulus and percentage of stress relaxation of GelMA hydrogels after 14 days in culture. (**a**) Elastic modulus of cell-loaded (ASC) and cell-free (Control) 5%, 10%, and 15% GelMA at a strain rate of 0.2 s^−1^ during the first second of compression. (**b**) Percentage of stress relaxation at constant strain constant over 100 s of cell-loaded and cell-free 5%, 10%, and 15% GelMA. (**c**) Average stress relaxation normalised to start from 100% for 100 s. Data derives from a minimum of five non-paired samples per timepoint, from three independent experiments. Data are presented as mean with standard deviation. *p* values are indicated * *p* < 0.05, ** *p* < 0.01, *** *p* < 0.001 and **** *p* < 0.0001 according to Two-way ANOVA and Tukey.

**Figure 4 ijms-22-10153-f004:**
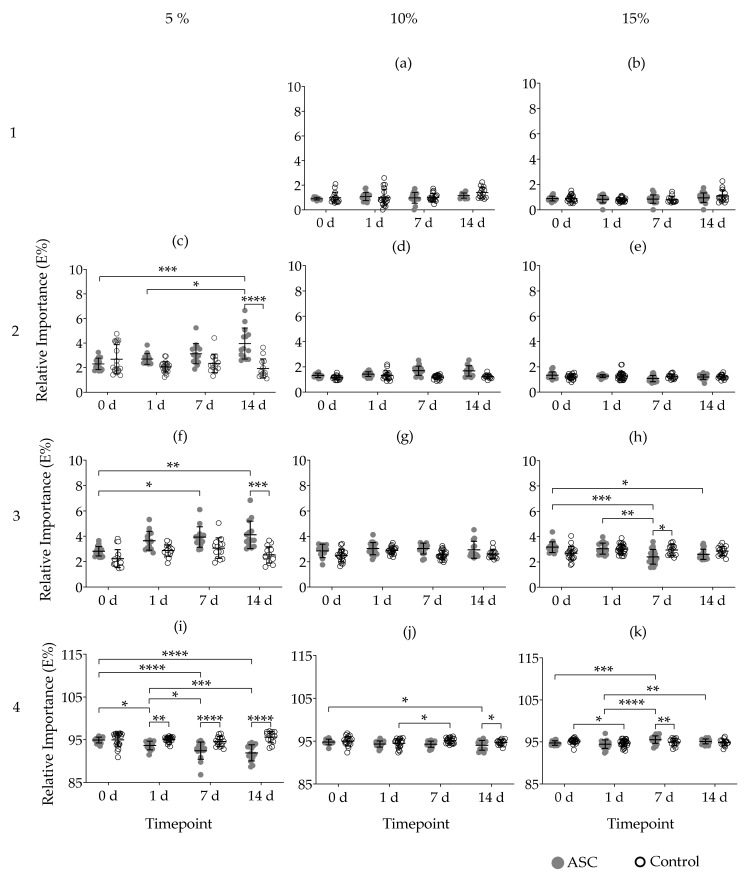
Relative importance of each Maxwell element of cell-loaded (ASC) and cell-free (Control) GelMA hydrogels after 14 days in culture. Fitting GelMA stress relaxation data into a generalised Maxwell model determined the number of Maxwell elements 1 (*τ* ≤ 1 s), 2 (1 < *τ* ≤ 10 s), 3 (10 < *τ* ≤ 100 s), and 4 (*τ* > 100 s) as well as the contribution of each element (relative importance or *R_i_*) to the stress relaxation process (**a**–**k**). Changes in *R_i_* of each Maxwell element in 5% (**c**,**f**,**i**), 10% (**a**,**d**,**g**,**j**), and 15% GelMA (**b**,**e**,**h**,**k**) are shown. Data derives from a minimum of five non-paired samples per timepoint, from three independent experiments. Data are presented as mean with standard deviation. p values are indicated * *p* < 0.05, ** *p* < 0.01, *** *p* < 0.001 and **** *p* < 0.0001 according to Two-way ANOVA and Tukey.

**Figure 5 ijms-22-10153-f005:**
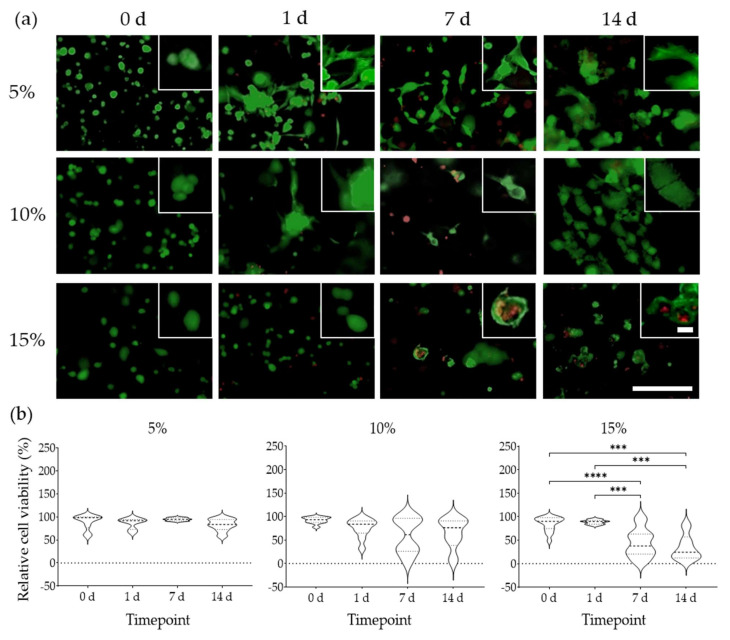
ASC morphology and viability in 5%, 10%, and 15% GelMA over 14 d. (**a**) Cells were stained with Calcein AM (green) and the nuclei of dead cells stained with PI (red). Scale bars depict 25 µm (magnified view) and 200 µm (landscape view). (**b**) Cell viability percentage was calculated based on ImageJ quantification of fluorescently labelled cells. Data derive from three independent experiments. Data are presented as median (dash line) and quartiles (dotted lines). *p* Values are indicated as *** *p* < 0.001 and **** *p* < 0.0001 according to Brown-Forsythe and Welch ANOVA and Dunnett´s T3 tests.

**Table 1 ijms-22-10153-t001:** Tau (*τ*) values in seconds (s) of each Maxwell element per GelMA concentration and timepoint.

	Cell-Loaded				Cell-Free			
	GelMA 5%							
Timepoint	*τ* _1_ ^1^	*τ* _2_ ^1^	*τ* _3_ ^1^	*τ* _4_ ^1^	*τ* _1_ ^1^	*τ* _2_ ^1^	*τ* _3_ ^1^	*τ* _4_ ^1^
0 d	-	1.52 ± 0.60	16.28 ± 3.98	3376.46 ± 828.45	-	1.26 ± 0.67	19.25 ± 9.56	4481.10 ± 2000.98
1 d	-	1.76 ± 0.66	20.56 ± 7.03	4240.52 ± 2191.19	-	1.54 ± 0.60	18.51 ± 5.06	4064.65 ± 792.60
7 d	-	1.08 ± 0.38	13.85 ± 3.39	3454.05 ± 786.02	-	1.40 ± 0.40	19.39 ± 4.84	4000.18 ± 1125.34
14 d	-	1.30 ± 0.32	14.60 ± 2.58	3477.60 ± 557.33	-	1.21 ± 0.55	16.90 ± 5.28	3551.34 ± 835.73
	**GelMA 10%**							
0 d	0.56 ± 0.18	4.43 ± 0.97	29.80 ± 3.80	3667.85 ± 461.86	0.45 ± 0.23	3.83 ± 0.96	26.19 ± 4.57	3888.82 ± 578.90
1 d	0.61 ± 0.26	4.39 ± 1.04	31.23 ± 7.20	5080.59 ± 1476.55	0.45 ± 0.25	4.61 ± 1.64	31.04 ± 6.39	5015.52 ± 738.22
7 d	0.52 ± 0.23	4.18 ± 1.45	27.44 ± 6.47	4770.59 ± 1445.05	0.37 ± 0.12	3.68 ± 0.97	27.22 ± 3.90	5108.56 ± 840.11
14 d	0.73 ± 0.68	4.15 ± 0.98	27.34 ± 4.91	5349.16 ± 1364.55	0.39 ± 0.16	3.62 ± 1.16	26.97 ± 8.54	5117.89 ± 1583.84
	**GelMA 15%**							
0 d	0.46 ± 0.19	4.27 ± 0.83	30.05 ± 4.11	3888.78 ± 663.64	0.58 ± 0.23	4.62 ± 1.50	31.16 ± 13.59	4595.73 ± 1612.28
1 d	0.56 ± 0.21	4.12 ± 1.49	28.16 ± 9.35	4095.32 ± 1304.40	0.60 ± 0.27	4.54 ± 1.50	30.62 ± 6.79	5477.63 ± 1708.10
7 d	0.45 ± 0.20	3.59 ± 1.12	27.35 ± 5.32	5280.01 ± 1752.85	0.49 ± 0.16	4.24 ± 0.61	30.30 ± 3.82	5185.26 ± 670.65
14 d	0.50 ± 0.20	3.91 ± 1.49	28.82 ± 7.95	6162.22 ± 1748.41	0.41 ± 0.15	3.90 ± 0.65	28.29 ± 2.93	4497.15 ± 462.33

^1^ Data in seconds. (-) No first Maxwell element was detected in 5% GelMA. All data is shown as Mean ± standard deviation (SD).

## Data Availability

The datasets generated during and/or analysed during the current study are available from the corresponding author upon reasonable request.

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
