# Peer review of "Adipose Tissue-Derived Stromal Cells Alter the Mechanical Stability and Viscoelastic Properties of Gelatine Methacryloyl Hydrogels"

_ijms, 2021, doi:10.3390/ijms221810153_

Round 1

Reviewer 1 Report

The authors evaluated changes in GelMA viscoelasticity and stiffness due to ASCs cultivation. The topic is interesting and the paper il well written, but I have some concerns:

1) Could you please reduce the exposure time of figure 5a? Some pictures seem to be overexposed, especially the ones showing the cells seeded in 5% and 10% gels. In addition, could you please add some inserts with higher magnification? They would allow the reader to better appreciate changes in the cellular morphology.

2) During the cell migration, what happened to the gel? Do the cells induces any additional changes in the gel structure?

3) In my opinion it would be more interesting and helpful to the readers if the authors could investigate in a deeply manner the mechanisms involved in the remodelling of the gel by the cells, elucidating what enzymes, proteins, molecules etc... are involved in the process.

Reviewer 2 Report

Hydrogels are commonly used for in vitro modeling of the extracellular matrix within which the cells reside. Using such a system, the authors have addressed an important question that has been less studied earlier i.e. how the cellular processes influence the viscoelastic microenvironments within the extracellular matrix. The authors have used adipose tissue-derived stromal cells (ASCs) loaded into gelatine methacryloyl (GelMA) hydrogels and mathematical modeling to show that ASCs modified hydrogel stiffness as well as transformed hydrogel viscoelasticity into a more fluid-like environment. The authors have properly defined the research question and performed rigorous investigation. Also, the conclusions and the limitations of the study are well stated. I would support the publication of this manuscript, given the following is addressed:

  1. The authors have used GelMA of various concentrations to assess the influence of ASC on the stiffness and viscoelasticity of GelMA. In the discussion section, it would be useful to add the physiological relevance of these concentrations of hydrogel in terms of its similarity to ECM and its usefulness for tissue engineering and regenerative medicine purposes.

Round 2

Reviewer 1 Report

The auhtors addressed the review's questions